# Punicalagin Regulates Apoptosis-Autophagy Switch via Modulation of Annexin A1 in Colorectal Cancer

**DOI:** 10.3390/nu12082430

**Published:** 2020-08-13

**Authors:** Thanusha Ganesan, Ajantha Sinniah, Zamri Chik, Mohammed Abdullah Alshawsh

**Affiliations:** Department of Pharmacology, Faculty of Medicine, University of Malaya, Kuala Lumpur 50603, Malaysia; thanushaganesan@gmail.com (T.G.); zamrichik@ummc.edu.my (Z.C.); alshaweshmam@um.edu.my (M.A.A.)

**Keywords:** pomegranate, annexin A1, colorectal neoplasms, apoptosis, autophagy

## Abstract

Punicalagin (PU), a polyphenol extracted from pomegranate (*Punica granatum*) husk is proven to have anti-cancer effects on different types of cancer including colorectal cancer (CRC). Its role in modulating endogenous protein as a means of eliciting its anti-cancer effects, however, has not been explored to date. Hence, this study aimed to investigate the role of PU in modulating the interplay between apoptosis and autophagy by regulating Annexin A1 (Anx-A1) expression in HCT 116 colorectal adenocarcinoma cells. In the study, selective cytotoxicity, pro-apoptotic, autophagic and Anx-A1 downregulating properties of PU were shown which indicate therapeutic potential that this polyphenol has against CRC. Autophagy flux analysis via flow cytometry showed significant autophagosomes degradation in treated cells, proving the involvement of autophagy. Proteome profiling of 35 different proteins in the presence and absence of Anx-A1 antagonists in PU-treated cells demonstrated a complex interplay that happens between apoptosis and autophagy that suggests the possible simultaneous induction and inhibition of these two cell death mechanisms by PU. Overall, this study suggests that PU induces autophagy while maintaining basal level of apoptosis as the main mechanisms of cytotoxicity via the modulation of Anx-A1 expression in HCT 116 cells, and thus has a promising translational potential.

## 1. Introduction

Colorectal cancer (CRC) is the third most common cancer worldwide [1] owing to the high mortality rates and increasing prevalence portfolio. In the US, 1.8 million new cases of CRC were recorded in 2018 [2]. The global burden of CRC is expected to increase by 60% by 2030 with 2.2 new million cases recorded and 1.1 million deaths [1]. CRC is a growing health concern both locally and globally and hence impinging a huge need to be addressed. Despite all this advancement in current treatment strategies for CRC, the metastatic progression and cytotoxicity effect of current chemotherapy drugs are yet to be resolved, therefore emphasizing an urgency to reveal effective alternative treatment schemes.

Punicalagin (PU), which is a part of the family of ellagitannins (ET), is known to be the largest polyphenol found in pomegranate (*Punica granatum*) husk with a molecular weight of 1084.71 g/mol [3]. It is held as the bioactive constituent responsible for more than 50% of pomegranate juice’s anti-oxidant properties [4]. PU has been shown to have multiple health benefits, including anti-cancer, cardio-protective [5] and neuro-protective [6] benefits. Among the challenges in administrating PU orally would be its not-yet-explored bioavailability and absorption profile. Punicalagin is reported to be broken down into different urolithins by gut microbiota, and they possess different anti-cancer effects [7]. Hence, optimization of PU delivery will be the next step to escalate the application to clinical settings upon confirmation of its anti-cancer effects in vitro and in vivo. The role of PU has especially been studied in multiple tumors—including gastrointestinal cancer [8], colon cancer [9] and thyroid carcinoma [10]—for its anti-oxidative, anti-inflammatory and pro-apoptotic properties; however, its role in modulating endogenous compounds and thus the crosstalk between apoptosis and autophagy has never been explored to date.

PU is commonly known to induce apoptosis in different types of cancer and only recently its role in autophagy and apoptosis-autophagy switch have been gaining the interest of researchers. In a study done on thyroid carcinoma, PU was discovered to induce apoptosis-independent cell death, autophagy. Treatment of PU did not affect the nuclear fragmentation, chromatin condensation or cell cycle distribution of BCPAP cells, but instead displayed increased LC3-II conversion, beclin-1 expression and p62 degradation [10]. Another study on the effects of PU treatment on human glioma cells, U87MG, revealed that this polyphenol is capable of activating both apoptosis and autophagy in eliciting its anti-cancer property [11]. These studies hence provided fundamental evidence of the possible involvement of both apoptosis and autophagy cell death mechanisms in the treatment of PU.

Hence, this study intends to explore the role of PU in modulating the interplay between apoptosis and autophagy by regulating Anx-A1, a glucocorticoid-derived protein, in colorectal cancer cells. 

## 2. Materials and Methods

### 2.1. Compounds

Punicalagin (PU) with purity ≥ 98% by HPLC was purchased from Sigma-Aldrich (St. Louis, MO, USA, Cas No. 65995-63-3), while 5-Fluorouracil with purity ≥ 99% by HPLC was obtained from Sigma-Aldrich (St. Louis, MO, USA, Cas No. 51-21-8).

### 2.2. Cell Culture

HCT 116 (colorectal carcinoma) and CCD 841 (normal colon epithelium) cells, obtained from ATCC were cultured in complete media which consists of Dulbecco’s Modified Eagle Medium (DMEM, Hyclone), 10% Fetal Bovine Serum (FBS, Hyclone) and 1% of 10,000 units/mL Penicillin and 10,000 µg/mL Streptomycin (Gibco). Cells were split when they reach 70%–90% confluency. Once the cells were confluent, the media were discarded and the bottom of the flask was washed with 2 mL of PBS twice. Then, 2 mL of Trypsin (Gibco) was added and incubated for 3 min to detach the cells from the bottom of the flask. A quantity of 3 mL of complete media was added to neutralize the trypsin. The cell suspension was then transferred to a 15 mL falcon tube using serological pipette and the tube was centrifuged at 350× *g* at 4 °C for 5 min. The supernatant was discarded and 1 mL of complete media was added to resuspend the cells. Then, 9 mL of complete media was added to T-75 flask and 1 mL of the cell suspension was added to the flask. Cells were incubated at 37 °C with 95% air and 5% CO_2_. The cells were maintained within 10 passages in consecutive experiments.

### 2.3. Cell Viability Assay

Cells were grown in 96-well plate at a density of 10,000 cells per well and treated with 9 different concentrations (0–100 µg/mL) of PU (Sigma-Aldrich, St. Louis, MO, USA) at three different time points; 24, 48 and 72 h. 5-Fluorouracil (Sigma-Aldrich, St. Louis, MO, USA) was used as a positive control and the cells were treated at similar time points and similar concentrations as PU (0–100 µg/mL). At the end of treatment period, 20 µL of 5 mg/mL MTT was added to each well and incubated for four hours at 37 °C with 95% air and 5% CO_2_. Then, the contents were discarded and 100 µL of DMSO was added into each well before the reading was taken at 570 nm using a microplate reader (Hidex, Finland). The experiment was conducted in triplicates. IC_50_ value of PU was determined for HCT 116 cells and the subsequent experiments were done using the IC_50_ value.

### 2.4. Cell Cycle Analysis by Flow Cytometry

Cells were seeded at the density of 200,000 cells per T-25 flask and treated with IC_50_ value of PU for 24, 48 and 72 h. Untreated cells were used as negative control. On experiment day (end of treatment day), 70%–80% ethanol and PBS were kept in the ice 1 h prior to running the assay. The cells were detached using trypsin and collected in 15 mL falcon tube. The tubes were centrifuged at 750× *g* at 4 °C for 5 min and they were washed with 1 mL cold PBS before being centrifuged again for 15 min at 750× *g*. Then, 500 µL of 70% ethanol was added to the cells stepwise while vortexing to avoid clumping and left overnight at 4 °C. The next day, the cells were then centrifuged for 15 min at 750× at 4 °C before being washed with cold PBS twice. Cells were calculated using haemoctytometer and 10^6^ cells were used for each test. Then, 500 µL of propidium iodide (PI) stain was added to the cells. The cells were kept on ice for 30 min and upon completion of incubation period, the cells were transferred to flow cytometry tube using a filter before injecting it through the flow cytometer (BD FACS Canto II, USA) to be analyzed.

Modifit software was used to analyze the results obtained to study the distribution of cells in different phases of cell cycle.

### 2.5. Annexin V by Flow Cytometry

Cells were seeded at the density of 200,000 cells per T-25 flask and treated with IC_50_ value of PU for 24, 48 and 72 h. The cells were detached using trypsin, collected in 15 mL falcon tube and centrifuged. The cells were washed using cold PBS and resuspended in 1× binding buffer at a concentration of 1 × 10^6^ cells/mL. Next, 100 µL of the cell suspension was transferred to a 5 mL culture tube and 5 µL of FITC Annexin V and 5 µL of PI were added into each tube. The cells were gently vortexed and incubated in the dark at room temperature for 15 min. A quantity of 400 µL of 1× binding buffer was added to each tube before being analyzed using a flow cytometer (BD FACS Canto II, USA).

### 2.6. Caspase 3/7, 8 and 9 Assay

Promega’s caspase kits (G8090, G8200 and G8210) were used to run the assay and the methods follow the protocol provided in the kits.

The cells were seeded at a density of 10,000 cells per well in a white-walled 96-well luminometer plate and treated with IC_50_ value of PU for 72 h. Before starting the assay, Caspase-Glo Reagent was prepared by mixing Caspase Glo substrate with Caspase Glo Buffer provided in the kit. The plate was removed from the incubator and it was allowed to equilibrate to room temperature. Then, 100 µL of Caspase Glo reagent was carefully added to each well and the plate was sealed with a lid. The contents were mixed on a plate shaker 300–500 rpm for 30 s. The contents were incubated for 30 min at room temperature. The luminescence of each sample was measured using a spectrophotometer.

### 2.7. Confocal Analysis

Cells were seeded at a density of 300,000 cells per well on a coverslip in a 6-well plate and treated with IC_50_ value of PU or PU+ CsH + WRW4 for 72 h. Cyclosporin H (CsH) and WRW4 antagonists are selective antagonists of the formyl peptide receptors (FPR). Treated cells were stained with MitoTracker red dye (life technology, CA, USA). Then, 50 µL of live staining solution was added to each well and incubated for 30 min at 37 °C. The medium was then aspirated and 100 µL of fixation solution was added to each well and the plate was incubated for 20 min at room temperature. The fixation solution was aspirated and 100 µL of 1× wash buffer was added to each well. Next, 100 µL of permeabilization buffer was added to every well and incubated for 10 min at room temperature and was protected from light. After washing twice with 100 µL of 1× wash buffer, 1× blocking buffer was added and left for 15 min at room temperature. Following that, 50 µL of cytochrome C primary antibody was added to each well and incubated for 60 min in dark. The plate was then washed three times with 1× wash buffer before 50 µL of DyLight 649-conjugated secondary antibody was added and the cells were incubated for another hour protected from light at room temperature. Hoechst 33342 blue dye (Thermo Scientific™, Pittsburgh, PA, USA) was added to stain the nuclei of the cells. Three washes with 1× wash buffer were repeated. After washing, the coverslip was transferred to a slide and sealed before viewing under the confocal microscope.

### 2.8. Proteome Profiler

A human apoptosis array kit from R&D Systems (Catalog no.: ARY009) was used for this assay. The experiment was conducted as per the instructor’s manual to test for 35 different apoptosis-autophagy related proteins.

### 2.9. ELISA for Annexin A1

Human Anx-A1 Elisa Kit from Elabscience (Cat No.: E-EL-H5512) was used. The experiment was conducted as per the instructor’s manual.

### 2.10. Autophagy Flux Assay

Manufacturer’s protocol was followed for CYTO-ID Autophagy Detection Kit 2.0 (Cat No.: ENZ-KIT175). Two applications were used to detect autophagy flux; flow cytometry and fluorescence microscopy.

### 2.11. Statistical Analysis

PRISM software (Version 6.0e) was used to analyze the results obtained from the experiments. One-way ANOVA, two-way ANOVA, and *t*-tests were used accordingly to interpret the results. Statistical significance was analyzed using Bonferroni Post-hoc test. Data were expressed as mean ± SEM and *p* value ≤ 0.05 considered significant.

## 3. Results

### 3.1. PU Exhibits Selective Cytotoxicity on HCT 116

The MTT results showed that PU exerts anti-proliferative effect and specific cytotoxicity on HCT 116, colorectal cancer cell line (Figure 1A); however, it does not induce significant cytotoxicity on normal colon cell line, CCD 841 (Figure 1B). The IC_50_ value of PU on HCT 116 was 87 ± 3.825 µg/mL (Figure 1A). Interestingly, 5-Fluorouracil, a common anti-cancer drug for colorectal cancer was observed to exert the same cytotoxicity trend on both HCT 116 (Figure 1C) and CCD 841 (Figure 1D); IC_50_ value of 5-FU on HCT 116 is 1.35 ± 0.03215 µg/mL.

### 3.2. PU Exhibits Apoptotic Properties on HCT 116

Cell cycle analysis showed that PU consistently caused significant a reduction in cell viability in S phase after 24, 48 and 72 h of treatment compared to the untreated control group, suggesting possible cell cycle arrest in PU-treated cells (Figure 2).

Annexin V assay demonstrated that in PU treated group, on average 11.9% ± 0.949% of cells entered early apoptosis compared to 4.3% ± 1.017% of untreated cells (*p* < 0.001) (Figure 3). There were no significant differences between treated and untreated group for late apoptosis (2.533% ± 0.731% vs. 2.467% ± 0.410%, respectively) and dead cells phases.

Confocal microscopy was used to investigate three different apoptotic parameters including DNA damage, mitochondrial membrane potential (MMP), and cytochrome C (Cyto C) release (Figure 4A). Treatment groups (PU-treated cells and PU + CsH + WRW4 (FPR inhibitors)) were compared with untreated control. FPR inhibitors are inhibitors of formyl peptide receptors, which are cognate receptors of Anx-A1. The inhibitors were used to identify the role that Anx-A1 was speculated to have in the progression of Anx-A1. The mean fluorescent intensity values quantified for MMP were significantly lower in PU-treated cells (3.563 ± 0.120) and PU + FPR inhibitors-treated cells (3.749 ± 0.224) compared to untreated cells (10.450 ± 0.496). On the other hand, the mean intensity measures for cytochrome C release were significantly higher in PU-treated cells (5.960 ± 0.524) and PU + FPR inhibitors-treated cells (10.717 ± 0.429) compared to untreated cells (4.202 ± 0.715) (Figure 4B). Based on the results, it was obvious that Anx-A1 indeed plays a role in cancer progression and it was plausible to further investigate its effect in PU-treated cells.

### 3.3. PU Downregulates Anx-A1 in HCT 116

ELISA results showed that treatment of PU on HCT 116 cells significantly reduced the expression of Anx-A1 (Figure 5A). In this experiment, we have utilized Anx-A1 receptor (FPR 1/FPR 2) inhibitors, cyclosporine H (CsH) and WRW4. Our data showed that FPR inhibitors increases the inhibition of cell viability compared to PU alone treated HCT 116 cells (Figure 5B). Inhibition of cell growth increased to 84.681% ± 1.359% (*p* < 0.001) compared to the control untreated cells. 

### 3.4. PU Exhibits Autophagic Properties on HCT 116

Flow cytometry analysis showed that autophagosome is significantly degraded in PU-treated HCT 116 cells when compared with chloroquine (CQ) treated HCT 116 cells, which explains lesser intensity of staining in the PU treated group. Figure 6A shows fluorescent image of autophagosomes in PU-treated cells and control (CQ treated cells). Chloroquine-treated cells have a higher intensity of staining of autophagosome compared to PU-treated cells, demonstrating the prominent degradation of autophagosomes. 

To complement the quantitative data, fluorescence microscopy was performed to obtain qualitative analysis on the concentration and distribution of autophagosome in the treated groups and the qualitative analysis matches with the flow cytometry analysis (Figure 6B).

Activity of caspase 3/7, caspase 8 and caspase 9 were downregulated at 72 h post-treatment with significant reduction in caspase 3/7 compared to the untreated control (Figure 7). This serves as an evidence of possible involvement of another cell death mechanism apart from apoptosis.

### 3.5. PU Regulates Apoptosis-Autophagy Interplay in HCT 116 Cells

Proteome profiler revealed that catalase, HSP 27 (heat shock protein), HSP 60 and TNFRI/ TNFRSF1A (tumor necrosis factor receptor 1/tumor necrosis factor receptor superfamily member 1A) are the four proteins that were significantly different between untreated and PU + FPR inhibitors-treated HCT 116 cells. Figure 8 shows the expression of the four significantly different proteins in the three different treatment groups. Amongst the 35 tested proteins, three proteins (HSP27, HSP60 and TNFR1/TNRSF1A) were significantly downregulated in the PU + FPR inhibitors-treated HCT 116 cells, while catalase was significantly upregulated as compared to untreated control.

## 4. Discussion

### 4.1. Anti-Proliferative and Apoptotic Effects of PU

Anti-proliferative and pro-apoptotic are major anti-cancer properties of PU which has been tested on multiple tumors in previous studies [4,12,13], so we intended to establish these properties in HCT 116 cells before achieving other aims. With the selective proliferative advantage being one of the seven hallmark characteristics of cancer [14], the anti-proliferative effect of PU is a crucial one to be established to determine its specific cytotoxicity on HCT 116 cells. 

MTT results demonstrated that PU shows an anti-proliferative effect and selective cytotoxicity on HCT 116, colorectal cancer cell line, as it does not impinge significant cytotoxicity on the normal colon cell line, CCD 841 (Figure 1A,B). This is an imperative finding, as current chemotherapeutic drugs for treatment of colorectal cancer do not conform to targeted therapy, which explains the side effects experienced by patients. This is demonstrated in our experiment whereby 5-Fluorouracil, a common chemotherapeutic drug for colorectal cancer, was found to exert a similar cytotoxicity trend on both cancer and normal cells (HCT 116 and CCD 841) (Figure 1C,D). The selectivity index (SI) of PU and 5-FU was calculated and PU had an SI of approximately 1.15, whilst 5-FU recorded a SI of 0.5. The comparatively higher SI of PU towards HCT 116 in comparison with 5-FU, demonstrates its selective cytotoxicity property on the cancer cells and cyto-protective effect on normal colon epithelium. A similar trend of cytotoxicity for PU on HCT 116 had previously been shown in studies [4]; however, its cytoprotective effect on normal colon epithelium was never reported. The anti-cancer effect of PU on cancer cells could possibly be explained by either its pro-oxidant or antioxidant properties. Whilst often the scavenging property of reactive oxygen species (ROS) is associated to anti-cancer property of polyphenols, the selective cytotoxicity could be due to the higher sensitivity of cancer cells to extra ROS produced by polyphenol [15].

Another important property of PU investigated in this project was the pro-apoptotic effect of the compound. One of the early features of apoptosis is the movement of phosphatidylserine (PS) of the phospholipid bilayer to the external surface and the externalization of PS is detected via conjugation of Annexin V to PS [16]. In the later stages of apoptosis, the membrane integrity can be compromised which allows the penetration and staining of PI dye. The data from Annexin V assay proved the apoptotic effect of PU as the percentage of treated HCT 116 that entered early apoptotic phase (11.933% ± 0.949%) was significantly higher than that of the untreated cells. However, only a small percentage of treated cells entered the late apoptotic phase (2.533% ± 0.731%) showing that the membrane integrity was possibly not compromised to a large extent in this treatment). Apoptosis can also be characterized by biochemical changes such as protein cleavage, protein cross-linking, DNA breakdown, loss of mitochondrial membrane potential and cytochrome C release [16]. Confocal analysis was used in the study to assess three parameters of biochemical changes that were possibly occurring following PU treatment on HCT 116 namely DNA damage, mitochondrial membrane potential and cytochrome C release. Qualitative analysis confirmed apoptotic effect of PU with higher cell death and more obvious DNA damage in the treated group compared to the untreated control. Quantitative analysis portrayed significantly decreased mitochondrial membrane potential and enhanced cytochrome C release in the treated groups (both PU alone and PU + FPR inhibitors) as compared to the untreated control. Mitochondrial membrane potential is inversely proportional to cytochrome C release, as compromise in the membrane potential will diminish cytochrome C release [17]. Hence, these two parameters are inter-dependent on each other.

Interestingly, PU + FPR inhibitors-treated HCT 116 had a more prominent decrease in MMP and higher cytochrome C release compared to PU alone treated group implying the possible role played by Anx-A1 in progression of CRC, which brings to the exploration of the second aim discussed in the following section.

To recapitulate, PU exhibits specific cytotoxicity, anti-proliferative effect and pro-apoptotic property and very promisingly has potential to be developed into a targeted therapy in the treatment of CRC.

### 4.2. Modulation of Anx-A1 by PU

Despite the role of PU being largely studied in different tumors, its activity in regulating an endogenous protein, especially for its anti-cancer mechanism was not largely explored and as for Anx-A1, this study is the first of its kind. Anx-A1 was of large interest for its role in inflammatory conditions, but lately interest is growing on its effect in cancer. Known to be differentially expressed in various types of cancers, Anx-A1 imposes multiple roles in cancer including but not limited to cellular proliferation, metastasis, drug resistance, migration, invasion and inflammation [18]. Thus, it was only reasonable to study the regulation of Anx-A1 expression by PU given the anti-inflammatory property of PU and the role this protein plays in inflammation.

Anx-A1 is known to be highly expressed in colorectal cancer [8]. We have shown that Anx-A1 expression is reduced in both HCT 116 and CCD 841 when treated with PU. The decrease in Anx-A1 expression was, however, more significant in HCT 116 compared to CCD 841, proving the selectivity that PU has towards CRC cell line. Higher expression of Anx-A1 in CRC in the past has been linked to lymphatic invasion, venous invasion, lymph node metastasis, advanced disease stages and lower disease-specific survival rate [8]. In another study, elevated Anx-A1 expression led to resistance of CRC cells to 5-FU, the main chemotherapy drug used to treat this cancer [18]. In accordance with that, the Anx-A1 ablation effect of PU could be speculated as one of its mechanism of actions in exerting its cytotoxicity on HCT 116. The mechanism by which Anx-A1 is inhibited by PU is not fully understood and hence, this opens up a new research vacuum to be explored and filled in the future. 

Upon discovering the role of PU in diminishing Anx-A1 expression in HCT 116, Anx-A1’s role in proliferation of HCT 116 was studied using Anx-A1 receptors, FPR 1 and FPR 2 inhibitors (cyclosporine H (CsH) and WRW4). CsH inhibits the Anx-A1/ FPR1 axis [19], whilst WRW4 inhibits the Anx-A1/ FPR2 axis [20], ensuring a more comprehensive inhibition of Anx-A1 and thus its actions. Following the inhibition, a significant decrease in cellular proliferation of HCT 116 was observed. A similar result was also observed in a breast cancer study when CsH and Cyclosporin A (CsA) were used to inhibit FPR1 in MDA-MB-231 cell line [19], portraying a universal role that Anx-A1 might have in ameliorating cancer cell proliferation and PU’s effect on reducing the expression of Anx-A1 in HCT 116 could thus be translated to other types of malignancies.

### 4.3. Autophagy-Apoptosis Switch and Anx-A1

Assessment on caspases involvement revealed a very interesting finding which steered the direction of the study. Possible involvement of another cell death mechanism was hypothesized for PU treatment following significant downregulation of caspases (caspase 3/7, 8 and 9) in PU treated cells as compared to the untreated control. 

Despite obvious cell death, caspase downregulation unravels induction of caspase independent cell death by PU. Based on literature evidences, PU was shown to induce autophagy in other types of cancer [11] and other physiological conditions [21] leading to possible speculation of the involvement of autophagy in PU treatment. Recapitulating all the results obtained, crosstalk between apoptosis and autophagy seemed possible in PU-treated HCT 116 and given that the pro-apoptotic property had been determined, involvement of autophagy was first determined before the interplay between these two programmed cell deaths were investigated.

Autophagy flux kit analysis via flow cytometry strengthens this speculation as the treated group has lesser autophagosomes than chloroquine (CQ), an autophagy inhibitor that causes accumulation of autophagosomes and prevents degradation of autophagosomes to complete autophagy [21]. Hence, the lower detection of autophagosome in the treated group indicates higher autophagy flux compared to the CQ treated cells as completion of autophagy involves the lysosomal degradation of autophagolysosome. Fluorescence microscopy imaging of autophagic vacuoles provided additional qualitative evidence for the induction of autophagy in the PU-treated group, given that the CQ group had denser staining of the vacuoles compared to the PU-treated HCT 116 cells. Combining autophagosomes’ quantitative analysis of flow cytometry and qualitative analysis of fluorescent images, PU’s pro-autophagic property was confirmed and the inter-connection between apoptosis and autophagy was sought to be explored in a condition where Anx-A1 is modulated in order to get a comprehensive picture of PU’s mechanism of action. 

A proteome profiling analysis which tested 35 different apoptosis-autophagy related proteins showed results supporting the theory of possible involvement of both the cell death mechanisms in treatment of PU in HCT 116 with the inhibition of Anx-A1 in place. The four proteins that were significantly different between the treated (PU and FPR inhibitors) and untreated control were HSP 27, HSP 60, catalase and TNF RI/ TNFRSF1A. Our results showed that both HSP 27 and HSP 60 were significantly decreased in the treated group compared to the untreated control, and it is worth taking note that the modulation of HSP 27 [22] and HSP 60 [23] are crucial for the induction of autophagy. On the other hand, catalase, which is often degraded in activation of autophagy, seemed to be upregulated in the PU and FPR inhibitors-treated groups. It is, however, proven in previous studies, that catalase degradation is driven by reactive oxygen species-stimulated autophagy and not starvation-induced autophagy [24]. This could possibly be a basis for speculation of a mechanism for the induction of autophagy in PU-treated HCT 116. TNF RI/ TNFRSF1A, a vital protein for extrinsic apoptotic pathway was significantly reduced in the treated group. A few other proteins that were not significantly different but quite remarkably modulated despite not capturing statistical significance and believed to play crucial role in apoptosis-autophagy crosstalk are HSP 70, Bax, p53, Bcl-2, Smac/Diablo, p21 and p27. All these proteins play roles in inducing or inhibiting either or both apoptosis and autophagy, reflecting a maze of complex and integrated signaling pathways.

HSP 70 inhibits apoptosis by stimulating anti-apoptotic protein Bcl-2/ Bcl-x and blocking Apaf-1, which is involved in activating caspase 3 when bound to caspase 9 [25]. In contradiction, Smac/ Diablo is a protein released upon mitochondrial damage and it is a caspase catalyzed event that occurs downstream to cytochrome C release [26]. 

On the other hand, p53 is a very crucial protein that has been implicated in many different cancers. This protein either induces or inhibits autophagy or apoptosis depending on its localization in the cell. p53 induces apoptosis regardless of its localization. In extrinsic apoptosis, p53 increases the expression of death receptors [27] and the TRAIL receptor [28], meanwhile caspase 3 and 8 are activated by cytoplasmic p53. On the other hand, in intrinsic apoptosis, nuclear p53 activates pro-apoptotic proteins such as PIDD, NOXA, Bax and Bid [29], leading to increased MMP, cytochrome C release and activation of caspase 9 [30], whilst cytoplasmic p53 translocates to mitochondria to bind to anti-apoptotic proteins to prevent them from binding to pro-apoptotic proteins, Bax and Bak [31]. Unlike apoptosis, in autophagy, p53 has contradicting roles depending on its localization. While cytoplasmic p53 activates the mTOR signaling pathway to inhibit autophagy [32], nuclear p53 induces transcriptional activity of damage-regulated autophagy modulator (DRAM) which is responsible for formation of autophagosome to activate autophagy [33]. Hence, this is a very interesting protein to explore apoptosis-autophagy interplay.

Protein p21 plays a crucial role in deciding the type of programmed cell death to be employed by the cell. In the presence of p21, autophagy related proteins might get degraded which allows induction of apoptosis demonstrating positive regulation of apoptosis and negative regulation of autophagy by this protein [34]. p27 on the other hand, exhibited functional differences based on its localization. Cytoplasmic p27 inclined towards enhancing autophagy while nuclear p27 promotes apoptosis and all in all, this protein is involved in modulating the balance of apoptosis-autophagy axis [35]. 

Analysis of the expression of these multiple proteins showed that there’s inclination towards both activation and inhibition of apoptosis and autophagy, therefore suggesting the existence of apoptosis-autophagy switch in PU treatment. A schematic diagram (Figure 9) was illustrated utilizing differently expressed proteins to show possible interplay between apoptosis and autophagy which could possibly explain the mechanism of action of PU in the presence of Anx-A1 inhibition.

In summary, our data suggest that PU treatment involves both apoptosis and autophagy in inducing cell death of colorectal cancer cells. PU exerts its anti-cancer effect by modulating apoptosis-autophagy switch via the downregulation of Anx-A1 protein in HCT 116 colorectal cancer cells. This finding provides a platform to develop therapeutic target of Anx-A1 to modulate different facets of cell death mechanism to specifically target tumor cells and protect the non-tumorigenic cells. 

The main translational challenge with delivery of PU to the tissues will be the breakdown of this compound into different metabolites which, in turn, will compromise its bioavailability. One in-vivo study reported that 3%–6% of PU or related metabolites were detected in the urine or feces, and the rest is believed to be converted to undetectable metabolites [36] indicating that the bioavailability of PU needs to be considered in clinical application. Microencapsulation is an effective method to encapsulate natural ingredients including polyphenol to protect them from nutritional and health loss [37]. Water and oil are usual materials used to create multi-layer encapsulation (for instance water-oil-water emulsion) to envelop the natural compound and preserve their bioavailability until reaching the target tissues [37]. One the other hand, in term of phytomedicine optimization, one of the tested ways to enhance the drug delivery of natural compound-derived drugs is to use nanoparticles [38]. Phytomedicines entrapped in nanocarriers possess a few advantages, such as higher therapeutic index, lower toxicity, lower dose and higher bioavailability [38]. Hence, upon confirmation of the anti-cancer property of PU in vivo in the future studies, the compound is worth being considered to be optimized either using microencapsulation or nanoparticle approaches to improve its bioavailability and delivery to the site of action to be qualified for clinical translation.

## 5. Conclusions

PU exerts its anti-cancer effect by modulating apoptosis-autophagy switch via downregulation of Anx-A1 protein in colorectal cancer cell line, HCT 116. It is a reasonable speculation that Anx-A1 could possibly be involved in deciding the direction of the cell death switch triggered by PU provided that the effect of PU on Anx-A1 expression was imminent. PU could be modulating the expression of Anx-A1 via inhibiting FPRs, as it was found out that PU’s anti-proliferative effect was potentiated in the presence of the FPR inhibitors, CsH and WRW4. PU treatment also seems to alter the expression of proteins involved in apoptosis and autophagy mechanisms, especially in the presence of FPR inhibition, demonstrating that it could have a role in transcription, translation and localization of these proteins. This finding provides a platform to develop therapeutic target of Anx-A1 to modulate different facets of cell death mechanisms to specifically target tumor cells and protect the non-tumorigenic cells. A more in-depth exploration on apoptosis-autophagy interplay in PU treatment in the future might also unravel new therapeutic targets in treatment of cancer.

## Figures and Tables

**Figure 1 nutrients-12-02430-f001:**
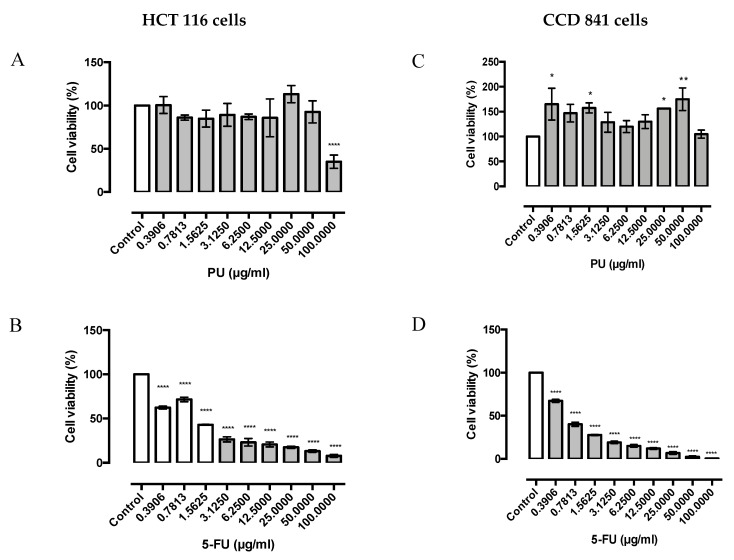
Punicalagin (PU) exhibits selective cytotoxicity on HCT 116 colorectal cancer cells as compared to CCD 841 normal colon cells. MTT assay for PU and 5-FU against HCT 116 cells (**A**,**B**) and CCD 841 cells (**C**,**D**) at 72 h. *, **, **** Significantly different from control (*p* < 0.05, *p* < 0.01, *p* < 0.0001); *n* = three independent experiments, PU: punicalagin, 5-FU: 5-fluorouracil.

**Figure 2 nutrients-12-02430-f002:**
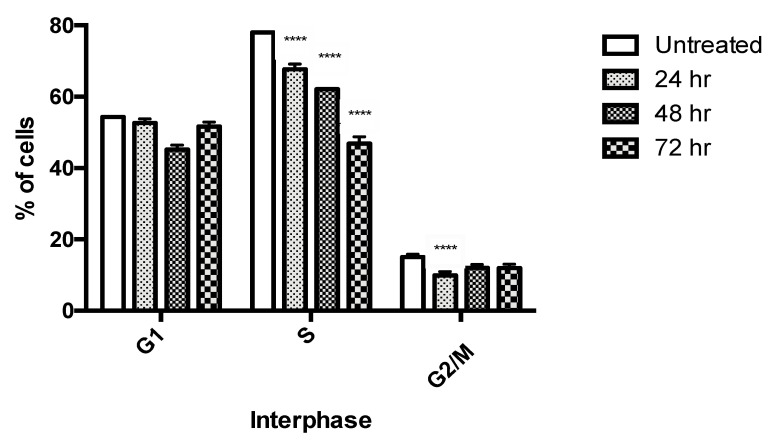
Cell cycle analysis of HCT 116 treated cells with punicalagin (PU) at 24, 48 and 72 h. PU treated HCT 116 was compared with untreated cells to study the progression through different phases of cell cycle. **** Significantly different from untreated control (*p* < 0.0001), *n* = three independent experiments.

**Figure 3 nutrients-12-02430-f003:**
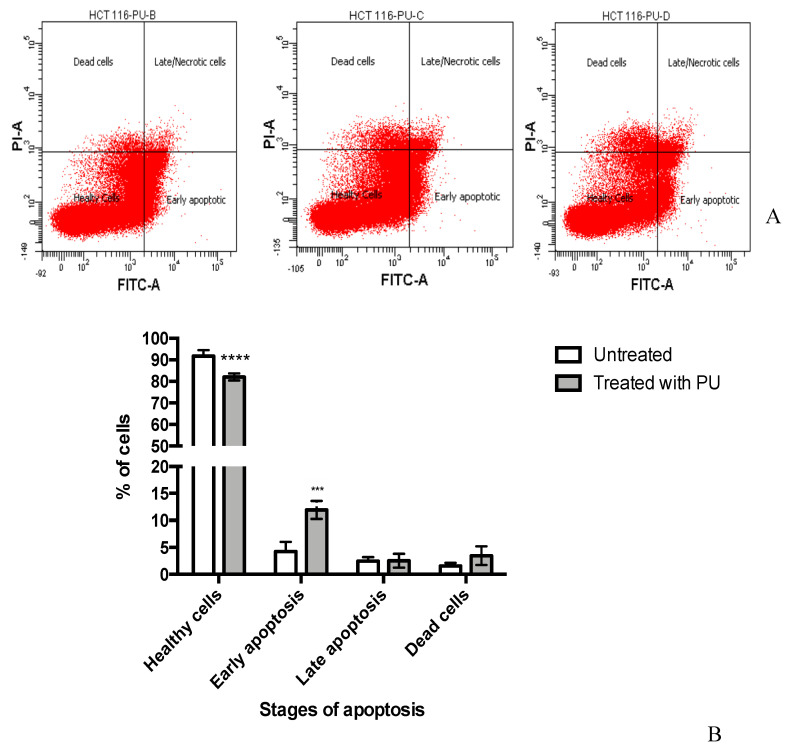
Punicalagin (PU) exhibits apoptotic properties on HCT 116 cells. (**A**) Annexin V assay of PU treated HCT 116 cells at 72 h to study distribution of the cells in different stages of apoptosis. (**B**) shows quantitative analysis of apoptosis stages. **** Significantly different from untreated control (*p* < 0.0001), *** (*p* < 0.001); *n* = 3 independent experiments.

**Figure 4 nutrients-12-02430-f004:**
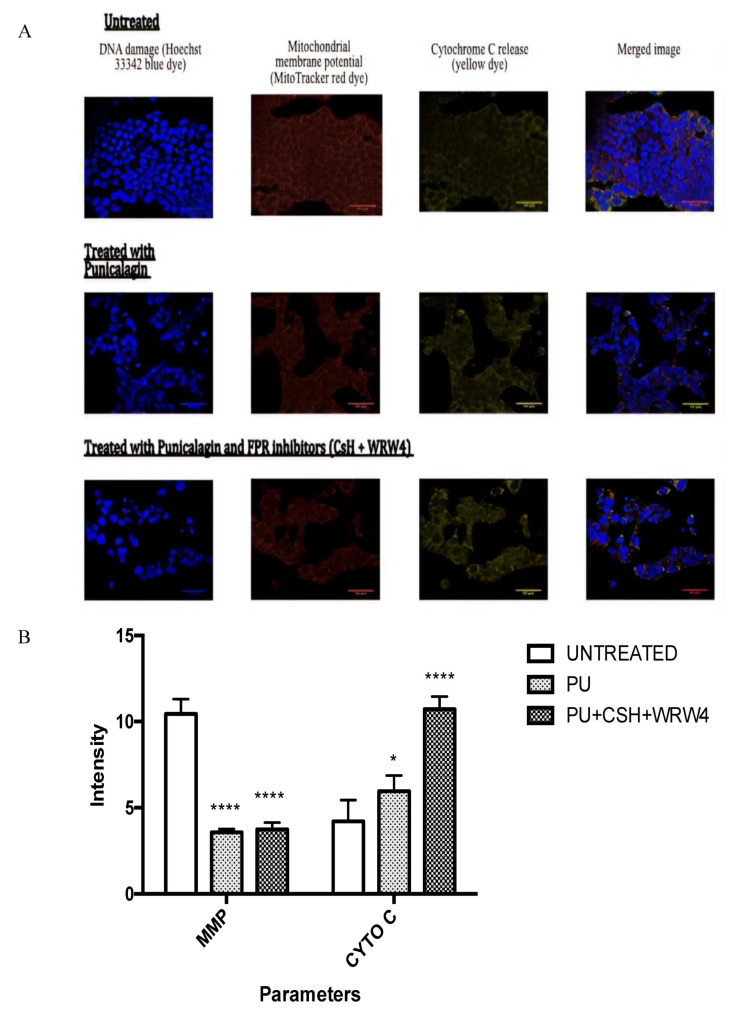
Confocal analysis of punicalagin (PU) treated HCT 116 cells at 72 h. (**A**) The effect of PU and PU + FPR inhibitors (CsH + WRW4) on DNA damage, MMP and cytochrome C release. Scale bars: 50 µm. (**B**) Shows quantitative analysis of the confocal images, **** Significantly different from untreated control (*p* < 0.0001), * (*p* < 0.05); *n* = three independent experiments. Cyto C: cytochrome C, MMP: Mitochondrial membrane potential, CsH: cyclosporin, FBR: formyl peptide receptors.

**Figure 5 nutrients-12-02430-f005:**
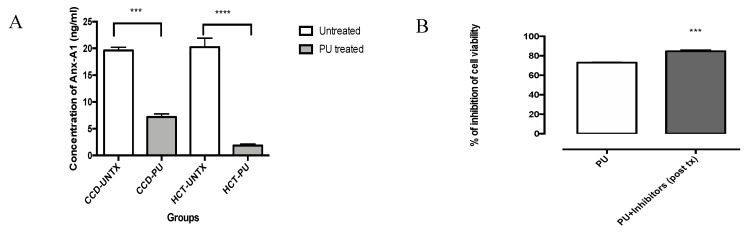
Punicalagin (PU) downregulates Anx-A1 in HCT 116 cells. ELISA was done on HCT 116 and CCD 841 cells at 72 h. (**A**) to compare downregulation of Anx-A1 in both cells lines in the presence and absence of PU treatment. (**B**) MTT assay was done on PU treated HCT 116 cells at 72 h and compared to PU + FPR inhibitors treated cells to study the role of Anx-A1 in cellular proliferation. **** Significantly different than control (*p* < 0.0001), *** (*p* < 0.001), *n* = three independent experiments.

**Figure 6 nutrients-12-02430-f006:**
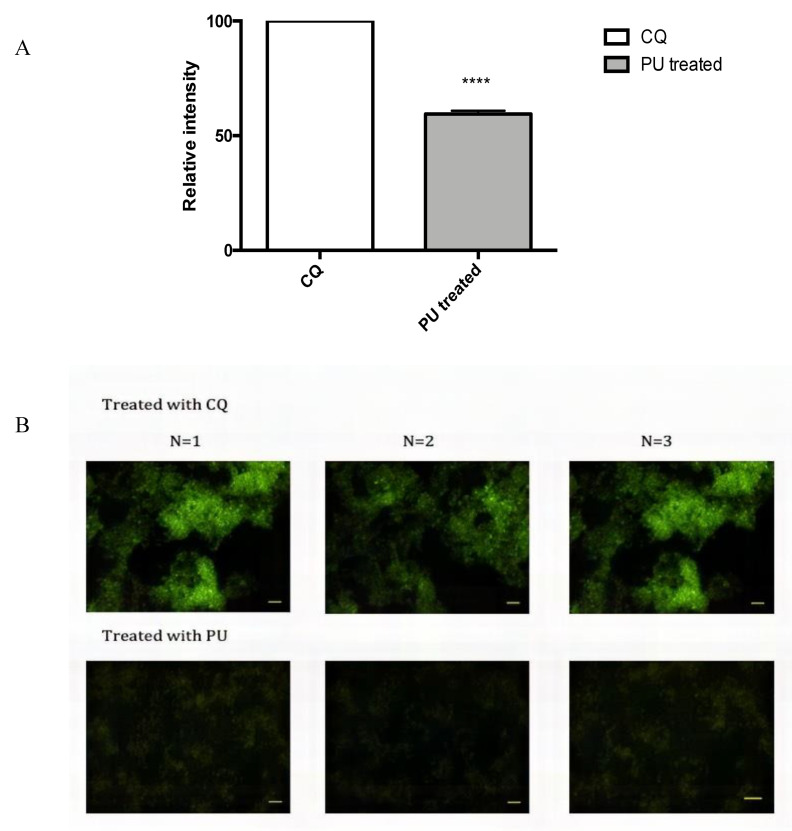
Punicalagin (PU) exhibits autophagic properties on HCT 116 cells. (**A**) Flow cytometry was used to analyze the density of autophagic vacuoles in PU treated HCT 116 cells at 72 h. Treated group was compared to chloroquine (CQ) treated HCT 116 cells as a negative control given that CQ is an autophagy inhibitor that causes accumulation of autophagosomes. (**B**) Qualitative analysis using fluorescence microscopy was done to complement flow cytometry analysis. Scale bars: 50 µm. **** Significantly different than chloroquine treated cells (*p* < 0.0001); *n* = three independent experiments.

**Figure 7 nutrients-12-02430-f007:**
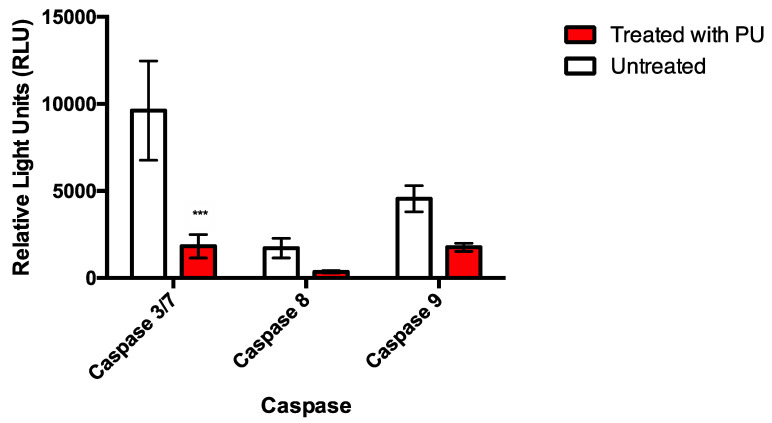
Caspase analysis was done on PU treated HCT 116 cells at 72 h and compared with untreated control cells to study expression of caspase 3/7, 8 and 9 in PU treatment. *** Significantly different than untreated cells (*p* < 0.001); *n* = three independent experiments.

**Figure 8 nutrients-12-02430-f008:**
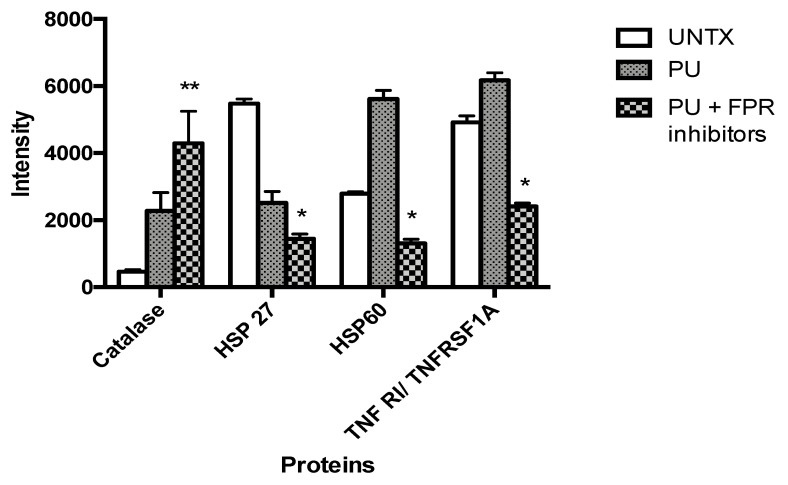
Punicalagin (PU) regulates apoptosis-autophagy interplay in HCT 116 cells via Anx-A1 modulation. Proteome profiling was done to study involvement of 35 different proteins in PU treatment on HCT 116 at 72 h in the presence and absence of Anx-A1 inhibitors (CsH and WRW4). The treatment groups were compared to untreated cells. ** Significantly different than control (*p* < 0.01), * (*p* < 0.05); *n* = three independent experiments.

**Figure 9 nutrients-12-02430-f009:**
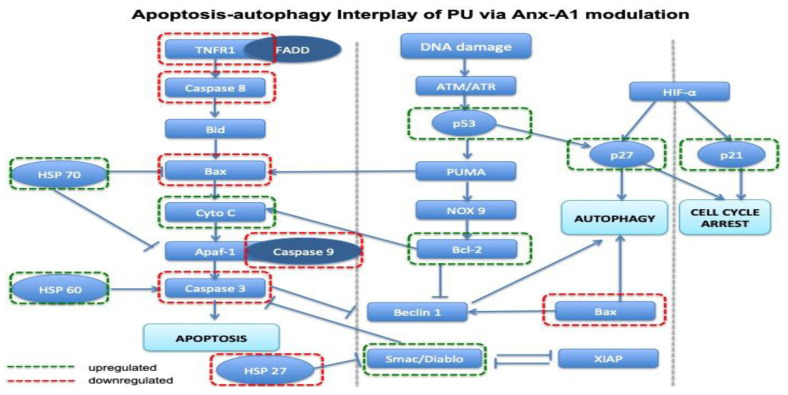
The cross talk between apoptosis and autophagy. Interplay between apoptosis and autophagy based on the proteins that were differentially expressed in the group treated with PU + FPR inhibitors compared to the untreated HCT 116 cells. Apaf-1: Apoptotic protease activating factor 1, ATM/ATR: Ataxia-telangiectasia-mutated/ Ataxia-telangiectasia and Rad-3, Bid: BH3 interacting-death domain agonist, Bcl-2: B-cell lymphoma 2, SP27: Heat shock protein 27, HSP 60: Heat shock protein 60, HSP 70: Heat shock protein 70, NOX: NADPH oxidase, PUMA: p53: Tumor protein p53, TNF RI: Tumor necrosis factor receptor 1, XIAP: X-linked inhibitor of apoptosis.

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
