# Peer review of "Punicalagin Regulates Apoptosis-Autophagy Switch via Modulation of Annexin A1 in Colorectal Cancer"

_nutrients, 2020, doi:10.3390/nu12082430_

Round 1

Reviewer 1 Report

Thank you for the possibility to review this manuscript. It covers a curious aspect with potential application character, especially for molecular biology oriented reader. The methodology is well and clearly described. Also discussion provides a concise and clear description of a core idea of the manuscript with confrontation of literature in the field. Authors try to explain possible practical applications of the study, what is undoubtedly a big strength of the manuscript, however, this compound is well known in the literature and it is not some outstanding news

Nevertheless, I have some comments to be addressed:

  1.  

Photos presented in manuscript are of a poor quality – please, attach in better resolution.

  1. Keywords should be searchable by the exact word/phrase in the MESH library (https://meshb.nlm.nih.gov/search it will improve visibility of the study in the internet and will be of benefit for future citations.
  2. please separate the test compound as a separate section not in connection with cell viability.
  3. please correct grammar and punctuation errors in manuscript (eg.in  2.1. 0Statistical Analysis).

Reviewer 2 Report

The manuscript entitled "Punicalagin regulates apoptosis-autophagy switch via modulation of Annexin A1 in colorectal cancer" needs to be revised. Some recommendations are as the following:

- Lines 70, 89, 92: Please convert rpm to g force.

- Line 77: The concentration of the 5-Fluorouracil applied on the cells is not given. Please clarify this issue.

- Line 148: Which other post-hoc tests are used besides t-test? Please clarify.

- Fig 1C: There are no stars above the bars. Are you sure that the treatments are not significantly different than the control? Please check.

- Fig 8: The following information is given in the figure legend: “n=3 independent experiments”. However, SD values are not shown?

- Line 284: “previous studies”. This information requires citation.

- Line 288-299: Polyphenols can act as either antioxidant or prooxidant, depending on the dose, cell type and cell culture conditions. In general, most bioactive actions are related to the reactive oxygen species (ROS) scavenging potential of the polyphenols. In contrast, their anticancer effect has been shown to be mediated through their prooxidant properties, as cancer cells have higher and more persistent oxidative stress levels compared to normal cells, which makes them more sensitive towards the extra ROS levels generated by pro-oxidants. This discussion may be inserted here by referring to the following paper: doi: 10.3390/nu7115462.

- Conclusion can be extended.

Reviewer 3 Report

The manuscript aims to evaluate the effects of punicalagin on apoptosis-autophagy processes via modulation of Annexin A1 in colon cancer cell model.

The manuscript presents two major lack in terms of methodology and background.

  1. Despite from a biological point of view it might be correct to set the concentration on the basis of the cell viability (as done by authors), authors should also take into consideration the in vivo applicability of the model, and care about the real physiological concentrations of the compounds once they reach the cells. Authors used a concentration of 87 μg/mL, which roughly corresponds to 80 μM, which is absolutely out of the physiological range. Again, despite it is appreciable to use cell as models and “increase” the concentration to see effects, this is not useful by thinking to real situations.
  2. Punicalagin is an ellagitannin which is deeply rearranged by gut microbiota strains, to different forms of urolithin (https://www.ncbi.nlm.nih.gov/pmc/articles/PMC3679724). For this reason punicalagin might not arrive to the cells or in very tiny concentration, while the presence of urolithin should be evaluate case by case due to the inter-individual variability.

Round 2

Reviewer 2 Report

Thank you very much for the revisions. The authors addressed almost all the comments that are raised by the reviewer. However, there is still one minor issue. I asked authors to convert rpm data to g force for the centrifugation processes. The authors responded to my comment as the following: “We could not get detail of the model of the centrifuge as the lab is closed due to the current lockdown, so we don’t have the rotor’s radius for conversion to g force”. I understand that under the current circumstances, it is not possible to go back to lab, however I find it quite strange that the authors did not ever mention the model of the centrifuge in any of their previous publications. Anyway, as I said, it is a very minor issue, but still strange.

Reviewer 3 Report

"the compound is worth for a consideration to be optimized using medicinal chemists’ expertise to enable it to qualify for clinical translation."

Here we are talking about foods! it is a Nutrition journal, not a medicinal chemistry one! You eat a food or a pomegranate extract and you see what arrives at cells. No one is talking about medicinal chemistry! You should stick to foods and how they affect health!

"It is true that punicalagin is broken down into different urolithins in the gut but there are different forms of urolithins (A, B, C, D) and they are all ought to exert different anti-cancer effects in colon cancer as reported by Gonzalez-Sarrias et al., [2]. So, it is not feasible to administer four different urolithins instead of a pure compound like punicalagin. In addition, punicalagin can be developed to be administered in parenteral or other form which will then not be affected by microbiota, which needs further pharmacokinetics and pharmacodynamics investigations, while our study is more to mechanistic investigation to confirm the anti-cancer effect of PU with focus on the apoptosis-autophagy switch. The bioavailability of punicalagin has been further discussed (line, 445-447, page 15) "

Authors are completely not matching the point. Despite in a future scenario you do not need to administer all the urolithins, but just the one which has activities (a company is selling pills of uro A), this cannot be say for punicalagin and more over at 87 uM concentration! And, more, here we are talking about foods and how the food components are affecting human health. If you are talking about administering punicalagin per IV to treat cancer, this is not the aim of the Journal and, more, you are just speculating this effect, as your cell model is not demonstrating any of this! If you give punicalagin to people IV you absolutely do not know if and how much of it arrives to colon cancer.
